

# Identification and validation of immune related core transcription factors *GTF2I* in NAFLD

Minbo Zhang[*], Yu Zhang[*], Xiaoxiao Jiao, Linying Lai, Yiting Qian, Bo Sun and Wenzhuo Yang

Department of Gastroenterology and Hepatology, Tongji Hospital of Tongji University, Shanghai, China
[*] These authors contributed equally to this work.

Corresponding authors
Bo Sun, sssunbo1995@tongji.edu.cn, sssunbo1995@163.com
Wenzhuo Yang, yangwenzhuo2002@163.com, 02891@tongji.edu.cn

## ABSTRACT

**Background**. Nonalcoholic fatty liver disease (NAFLD) is the most common liver disease worldwide that endangers human health. Transcription factors (TFs) have gradually become hot spots for drug development in NAFLD for their impacts on metabolism. However, the specific TFs that regulate immune response in the development of NAFLD is not clear. This study aimed to investigate the TFs involved in the immune response of NAFLD and provide novel targets for drug development.
**Methods**. Microarray data were obtained from liver samples from 26 normal volunteers and 109 NAFLD patients using the Gene Expression Omnibus (GEO) database. Differentially expressed genes (DEGs) were analyzed by limma package. Differentially expressed transcription factors (DETFs) were obtained on DEGs combined with Cistrome Cancer database. Immune signatures and pathways hallmark were identified by ssGSSEA and GSVA. The co-regulation network was constructed by the above results. Further, quantitative Real-time Polymerase Chain Reaction (qRT-PCR), Western blot (WB) and Immunohistochemistry (IHC) were used to validate the relationship between *GTF2I* and NAFLD. CIBERSORT analysis was performed to identify cell types to explore the relationship between differential expression of *GTF2I* and immune cell surface markers.
**Results**. A total of 617 DEGs and six DETFs (*ESR1*, *CHD2*, *GTF2I*, *EGR1*, *HCFC1*, *SP2*) were obtained by differential analysis. Immune signatures and pathway hallmarks were identified by ssGSSEA and GSVA. *GTF2I* and *CHD2* were screened through the co-regulatory networks of DEGs, DETFs, immune signatures and pathway hallmarks. Furthermore, qRT-PCR, WB and IHC indicated that *GTF2I* but not *CHD2* was significantly upregulated in NAFLD. Finally, *in silico*, our data confirmed that GTF2I has a wide impact on the immune profile by negatively regulating the expression of the chemokine receptor family (227/261, count of significance).
**Conclusion**. *GTF2I* plays a role in NAFLD by negatively regulating the chemokine receptor family, which affects the immune profile. This study may provide a potential target for the diagnosis or therapy of NAFLD.

## INTRODUCTION

Non-alcoholic fatty liver disease (NAFLD) is a complex metabolic disorder characterized by the accumulation of fats mainly with the form of triglycerides in hepatocytes more than 5% (*Brunt et al., 2015*). NAFLD covers a histological spectrum of liver diseases ranging from simple steatosis to non-alcoholic steatohepatitis (NASH), and subsequently can lead to fibrosis, cirrhosis, and even liver cancer (*Stefan, Häring & Cusi, 2019*). The prevalence of NAFLD has been reported to be 25% in adults worldwide and it has become the most prevalent causes of chronic liver disease in China (*GBD, 2020*; *Estes et al., 2018a*; *Estes et al., 2018b*). NAFLD, especially after progression to NASH, largely endangers human health and imposed significant burden on patients in terms of quality of life and economy (*Gordon et al., 2020*; *Younossi et al., 2016*). Despite the high burden of NAFLD on the community, the molecular and cellular mechanisms involved in NAFLD have not been fully understood and there is no approved drug regimen so far (*Younossi, 2019*; *Zhou et al., 2019*). Therefore, it is urgent to explore the underlying molecules in the pathogenesis and development of NAFLD.

The mechanisms underlying NAFLD are not entirely understood, the role of immune response in NAFLD has been the focus of intense research in the past few years (*Marchisello et al., 2019*). Innate and adaptive immune cell activation together with oxidative stress, mitochondrial and ER dysfunctions lead to necro-inflammation thus promoting NAFLD development (*Dong et al., 2007*; *Gebru et al., 2021*; *Suppli et al., 2019*). Transcription factors (TFs), which are frequently aberrant in diseases, have roles at focal points in signaling pathways, controlling many normal cellular processes such as cell growth and proliferation, metabolism, apoptosis, immune responses, and differentiation (*Lambert et al., 2018*; *Deng et al., 2022*). Nowadays, TFs are gaining increasing attention as drug targets against many diseases including NAFLD (*Papavassiliou & Papavassiliou, 2016*; *Becskei, 2020*). Studies have shown that hepatic de novo lipogenesis is mainly regulated by TFs, such as sterol regulatory element binding protein-1c (*SREBP-1c*), carbohydrate response element binding protein (*ChREBP*), Farnesoid X receptor (*FXR*), and peroxisome proliferator-activated receptor (*PPAR*) (*Ahmed & Byrne, 2007*; *Zhao et al., 2020*; *Hong & Tontonoz, 2008*; *Schmuth et al., 2014*; *Moon, 2017*). Transcription factor nuclear factor erythroid 2-related factor 2 (*Nrf2*) mediates the crosstalk between lipid metabolism and antioxidant defense mechanisms in experimental models of NAFLD (*Chambel, Santos-Gonçalves & Duarte, 2015*). Forkhead box O1 (*FOXO1*) plays a crucial role in coordinating the nutritional signals regulating metabolic control, including the homeostasis of glucose and lipids, inflammation, and oxidative stress (*Sabir et al., 2022*). Most of studies focus on roles of TFs in the process of metabolism; few are focusing on the effects of TFs on immune response, which plays major role in the progression of NAFLD (*Schmuth et al., 2014*; *Toobian, Ghosh & Katkar, 2021*; *Porcuna, Mínguez-Martínez & Ricote, 2021*). Therefore, it is necessary to explore the key TFs that regulate immune response and construct the regulation network of them in NAFLD. The key TFs could be the potential targets of NAFLD.

In the present study, we integrated the NAFLD population gene information, including pathway hallmarks, immune signatures, differential transcription factors and differential genes, to construct a co-regulatory network. Here, the hub gene-*GTF2I* was identified in this network. Furthermore, *in vitro*, in silico, and in human samples, our data confirmed that *GTF2I* may play a role in NAFLD by influencing the immune profile through negative regulation of chemokine receptor family. Our work may provide new insights of *GTF2I* involved in NAFLD and provide new target for the diagnosis and treatment of NAFLD.

## MATERIALS & METHODS

### Patients

Liver tissues used in the study were obtained from patients with hepatic hemangioma who underwent hepatic hemangioma resection in Tongji Hospital (Shanghai, China). The use of specimens was approved by the Research Ethics Committee of Tongji Hospital, and the Institutional Review Board approval number is K-KYSB-2020-139. All patients signed an informed consent to participate in this research.

### Cell culture and hepatic steatosis model constructed

At the condition of 37 °C, 5%CO2, HepG2 (HB-8065, ATCC) was cultured in 6-well plates ($2*10^5$ cells/well) of DMEM Medium (KGM12800-500, Keygene BioTECH, Nanjing, China) supplemented with 10% FBS (Gibco, South America) and 1% penicillin-streptomycin solutions. Cells were starved for 6 h in serum-free DMEM, 0.5% BSA (Sigma-Aldrich, St. Louis, MO, USA). Bovine albumin (BSA, 1mM) and free fatty acids (FFA,1mM) were involved in the medium for 24 h to induce control/steatosis cells (*Kanuri & Bergheim, 2013*; *Chavez-Tapia, Rosso & Tiribelli, 2011*).

### Data acquisition

The inclusion and exclusion criteria of participants were as described in previous studies. Specifically, microarray data and clinical information of 26 liver samples from healthy participants and 109 liver samples from NAFLD patients were achieved from the Gene Expression Omnibus (GEO) database (accession number: GSE126848, GSE130970) (*Brenner et al., 2013*; *Sun & Gao, 2004*). The participants included in this analysis were all diagnosed with histologically normal and NAFLD liver by biopsy, and interference from excessive alcohol consumption, diabetes mellitus, drug injury and other factors were excluded. In addition, the annotation files were acquired from GPL18573 Illumina NextSeq 500 (Homo sapiens) and GPL16791 Illumina HiSeq 2500 (Homo sapiens) platforms (*Hoang et al., 2019*; *Suppli et al., 2019*). Hallmark signaling pathways were collected from the Molecular Signatures Database. 29 immune-associated gene sets which represented diverse immune cell types, functions, and pathways were used to identify immune signatures (*He et al., 2018*).

### Data prerecession and differential expression analysis

Participants with incomplete samples and clinical information were excluded. Affy package was utilized for reading original microarray data. Robust multi-array average (RMA)

background was to correct, standardize, probe-specific background correction, and summarize probe set values in expression measure. Meanwhile, normalization was used to correct the integrated microarray data in preparation for differential expression analysis. Differential expression analysis of genes between NAFLD and normal liver were conducted using the Linear Models for Microarray Data (limma) package. False discovery rate (FDR) was utilized for multiple testing. FDR < 0.05 as well as absolute $\log_2$[Fold Change (FC)] $\geq$ 1.0 were cut off criteria.

## Functional enrichment analysis

Gene Ontology (GO) and Kyoto Encyclopedia of Genes & Genomes (KEGG) enrichment analyses were carried on to investigate intrinsic biological processes and pathways which were significantly related to DEGs with FDR *p*-value < 0.05 as cut off value. Furthermore, we obtained all transcription factors (TFs) from the Cistrome Cancer database (http://cistrome.org/CistromeCancer/), which were then merged with DEGs to acquire the key differentially expressed transcription factors (DETFs).

## Identification of potential downstream hallmark pathways

Potential downstream hallmark pathways of DETGs were identified through GSVA and GSEA (*Ferreira et al., 2021*; *Mei et al., 2017*). 50 hallmark pathways (GSEA | MSigDB (gsea-msigdb.org)) were absolutely quantified and then evaluated to extract the hallmark differential expression pathways related to NAFLD through ClusterProfiler package and GSVA package (*Subramanian et al., 2007*; *Yu et al., 2012*). Significant enrichment of upregulated and downregulated hallmark pathways in NAFLD and normal livers was indication through GSEA.

## Regulation network of transcription factors

Key DETFs ultimately associated with NAFLD were obtained through combing the Pearson correlation analysis of DETFs (correlation more than 0.85; significance less than 0.05), merging with the results of GSVA and GSEA analysis, and then plotted the regulatory network combining TFs, DEGs, and hallmark pathways using the igraph R package (https://igraph.org/r/).

## Oil red O staining

After HepG2 was treated with FFA for 24 h, the medium was removed. Wash twice with phosphate buffered saline (PBS) and fix with 4% paraformaldehyde for 30 min. Then, add oil red O (0.6% oil red O in isopropanol: $H_2O$ = 3:2) for 1 h and washed with PBS three times. The formation of lipid droplets was observed under optical microscope.

## Hematoxylin-eosin staining (HE) and immunohistochemistry (IHC)

Hematoxylin-eosin Staining (HE) was performed on liver tissue (5 mm) of three healthy subjects and 10 patients with NAFLD. Following routine rehydration, antigen retrieval, and blocking procedures, the sections were incubated overnight with *GTF2I* antibody (1:50 dilution, ab248269, Abcam, Cambridge, UK) at 4 °C. All slides were labeled polymer HRP for half an hour and hematoxylin as a counterstain for 5 min at room temperature.

## Western blot

30 mg of each sample was taken and lysed with RIPA at 4 °C to obtain the total protein. After protein quantification using a microplate reader, an appropriate amount of loading buffer was added and subsequently changed by heating in 95 °C for 15 min. After polyacrylamide gel electrophoresis (30 g protein usage) and subsequent routine western blot experimental steps, *GAPDH* (diluted 1:25,000, 60004-1, Proteintech, USA), *CHD2* (diluted 1:1,000, 12311-1-AP, Proteintech, USA) were respectively incubated overnight with the corresponding PVDF membrane region with the *GTF2I* (1:1,000, ab248269, abcam, USA) antibody at 4 °C overnight. Then, the membranes were incubated with the horseradish peroxidase-conjugated goat anti-rabbit IgG secondary antibody (cat. no. A0208; 1:10,000; Beyotime Institute of Biotechnology, Jiangsu, China) or goat anti-mouse IgG (cat. no. A0216; 1:10,000; Beyotime Institute of Biotechnology) at room temperature for 1 h. BeyoECL Plus (Beyotime Institute of Biotechnology, Jiangsu, China) and QuantityOne v4.6.6 software (Bio-Rad Laboratories, Inc., Hercules, CA, USA) were used to observe the protein bands.

## Quantitative reverse transcriptase-PCR (qRT-PCR)

Total RNA of liver specimens (30 mg) was extracted *via* Trizol reagent as described by the manufacturers, and RT-PCR were performed with total RNA (1 μg) according to the instructions of the RT-PCR reaction kit. The product was amplified in a reaction volume of 20 μl containing 1 μl RT product, 10 μl SYBR H (5X), and 0.5 μl of each primer (10X). PCRs were reacted for 40 cycles at 95 °C for 10 s, 60 °C for 30 s, 72 °C for 30 s. Samples' threshold cycle values were standardized to *GAPDH* mRNA expression, while the fold-change for each mRNA was calculated using the 2-delta delta Ct method. Primers of qRT-PCR were listed in Table 1.

## Immune profile assessment

According to the median of *GTF2I* in all samples, the samples were divided into high expression group and low expression group. CIBERSORT algorithm was used to analyze the effect of *GTF2I* on different immune cell composition. GSEA was used to analyze the differential pathways between high expression group and low expression group. Immune cell surface markers were used to further compare the difference of high expression group and low expression group.

## Statistics analysis

Statistical analyses were performed with R version 3.6.1 (*R Core Team, 2019*). The continuous variables were presented as the mean ± standard deviation. For non-microarray data, data that satisfied normal distribution were analyzed by Student's *t*-test. A Mann–Whitney *U* test was used to evaluate non-parametric data. *P* value < 0.05 was supposed to statistically significant.

**Table 1  The primer sequences.**

|  | Forward | Reverse |
|---|---|---|
| *GTF2I* | TTGTCGTCGGAACTGAAAGAG | CGATTTGCCTGGGTTGTAGAT |
| *CHD2* | AGTCAGTCGGAAAGTGAGCAG | ACATCAGCTATCCGTTCCTTCT |
| *GAPDH* | AATGGGCAGCCGTTAGGAAA | GCGCCCAATACGACCAAATC |

## RESULTS

### Identification of DEGs and functional enrichment analysis

The analysis procedure of this study is shown in Fig. 1. The participants included in this analysis were all diagnosed with histologically normal and NAFLD liver by biopsy, and interference from excessive alcohol consumption, diabetes mellitus, drug injury and other factors were excluded. On this basis, we integrated the expression matrix of the microarray data for differential gene analysis. As shown in the volcano plot (Fig. 2A) and heat map (Fig. 2B), 617 genes were identified as DEGs from 18435 genes in 26 controls and 109 NAFLD patients.

Next, GO and KEGG enrichment analyses were used to explore the potential function of identified DEGs. In GO analysis, the most significant terms of biological process (BP), cellular components (CC), and molecular function (MF) were 'establishment of protein localization to membrane' (Fig. 2C), 'cytosolic ribosome' (Fig. 2D), and 'structural constituent of ribosome' (Fig. 2E), respectively, which were critical in NAFLD pathogenesis. In KEGG analysis, DEGs between NAFLD and normal participants were involved in cell signal transduction and immune related diseases, such as 'Glycine, serine and threonine metabolism', 'Cell adhesion molecules', 'autoimmune thyroid disease' and 'Allograft rejection' (Fig. 2F). Through the functional analysis of differential genes between NAFLD and normal participants, these results suggested that the expression pattern of differential genes in NAFLD may be closely related to the immune profile, which we will investigate in the next step.

### Identification of key differentially expressed transcription factors (DETFs)

The results of the analysis above showed that differential genes were enriched in immunity and signal transduction. We further screened transcription factors on the basis of DEGs to obtain key regulatory genes.

All TFs were obtained through the Cistrome Cancer database, and then crossed with the DEGs to get the relevant transcription factors that are differentially expressed. As the volcano plot (Fig. 3A) and heat map (Fig. 3B) illustrated, six TFs (*ESR1, CHD2, GTF2I, EGR1, HCFC1, SP2*) were considered as the DETFs in patients with NAFLD. Among these transcription factors, *CHD2, ESR1*, and *GTF2I* were significantly increased in NAFLD. *EGR1, HCFC1* and *SP2* were significantly decreased in NAFLD. These transcription factors would prepare for the later construction of co-expression networks among immune signatures, pathway hallmarks, DEGs and DETFs.

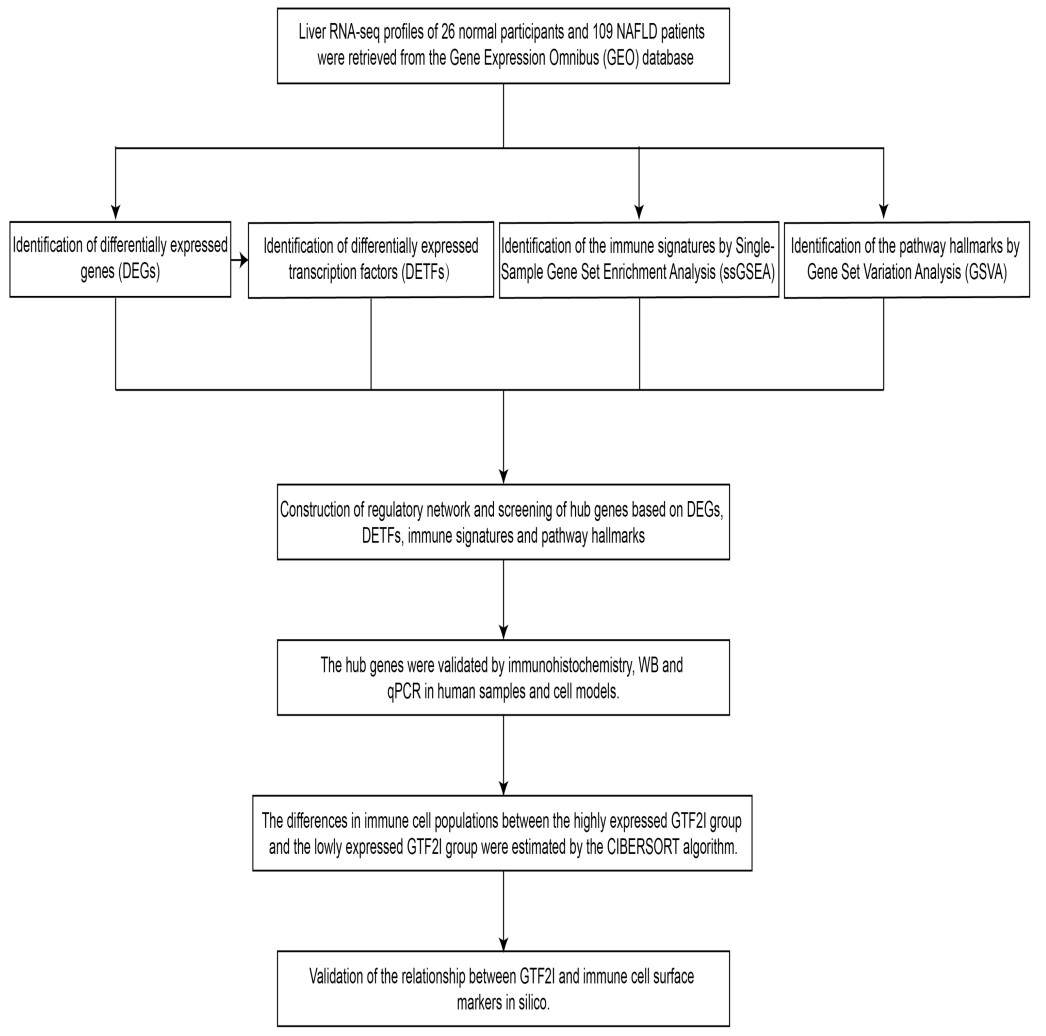

**Figure 1** Analysis flow chart of the study process of this study.

## Immune signatures and pathway hallmarks between NAFLD and normal participants

Based on the enrichment analysis results, we further deeply evaluated the distinction of immune profile in NAFLD and normal participants. 29 immune-associated gene sets which represented diverse immune cell types, functions, and pathways were used to identify immune signatures between NAFLD and normal participants. Firstly, we used the ssGSEA algorithm to rank individual sample gene expression by absolute value, compute cumulative empirical probability distributions for specific gene sets, and finally obtain enrichment score (ES) representing the abundance of immune infiltrating cells. The results were shown in the heatmap (Fig. 4A). and the most significant immune related gene set based on difference analysis were 'APC co-inhibition', 'DCs', 'Cytolytic activity', 'T helper cells' and 'CD8+ T cell' (Table 2).

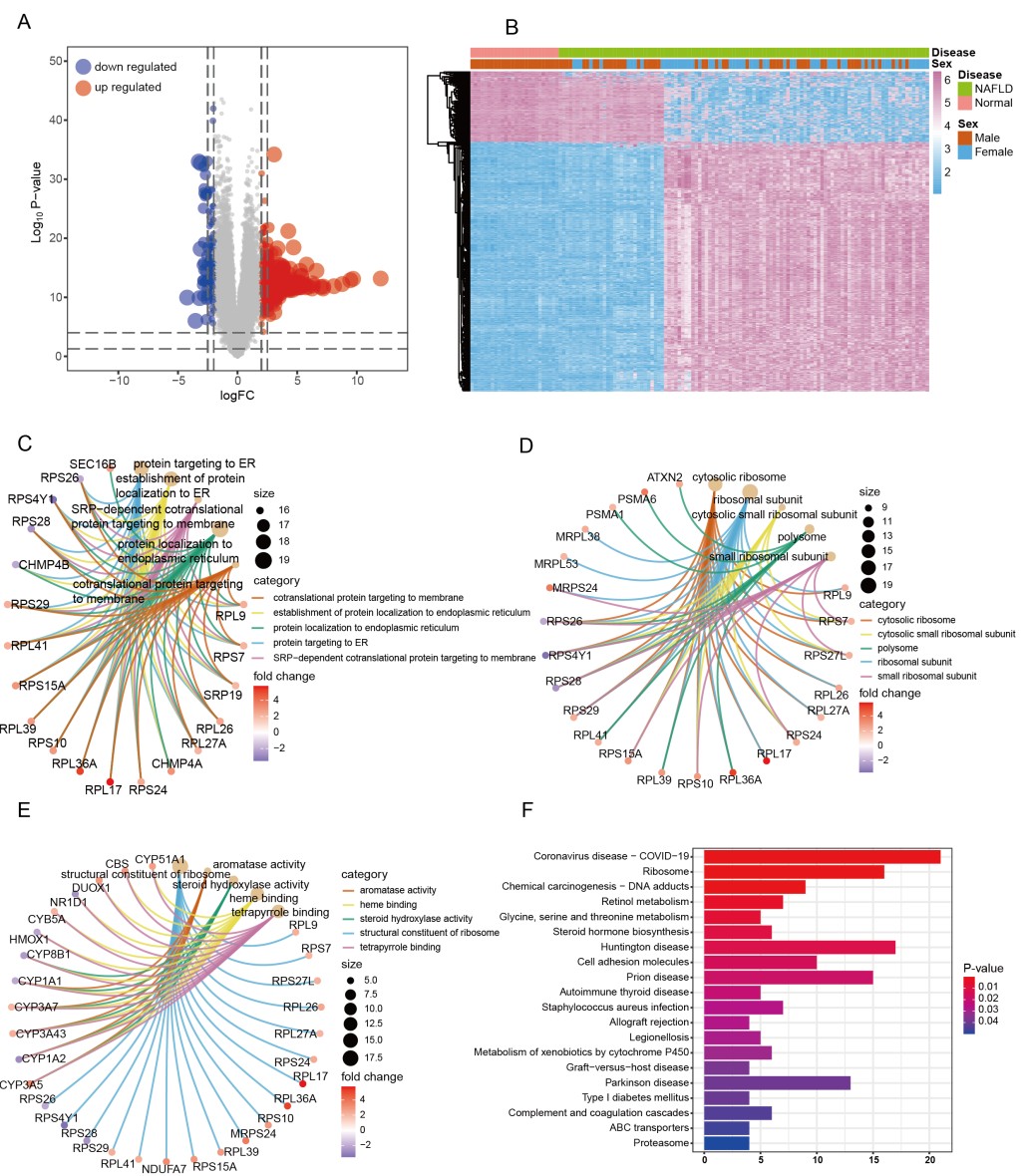

**Figure 2 Identification of differentially expressed genes (DEGs) and functional enrichment analysis.** (A) A total of 617 genes were identified as DEGs and visualized by volcanic map. (B) The expression levels of DEGs in each sample were visualized by heat map with unsupervised clustering. (C–E) The top five terms of BP, CC, and MF in GO analysis were visualized by circle plot. (F) The top 20 terms in KEGG analysis were visualized by bar plot.

In addition, at the biological function rather than a single gene level, we used the GSVA algorithm to analyze the biological status between NAFLD and normal participants to identify key biological processes. Hallmark gene sets, which summarize and represent specific well-defined biological states or processes, were used as reference datasets. The difference analysis results of GSVA in the pathway between NAFLD and normal participants were visualized by volcano map and heat map (Figs. 4B–4C). According to the differential
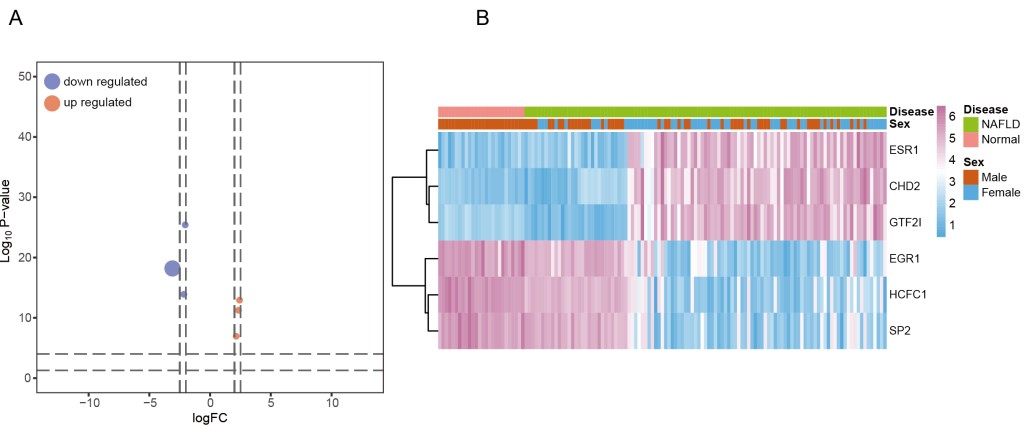

**Figure 3 Recognition of key DETFs and functional annotation.** (A) Six genes were identified as DETFs and visualized by volcano plot. (B) The expression levels of DETFs in each sample were visualized by heat map with unsupervised clustering.

ranking of each gene set between the two groups, the most significant gene sets were 'E2F_targets' and 'ANGIOGENESIS', which play an important role in cell proliferation and metabolism (Fig. 4D). The results of ssGSEA and GSVA analyses helped us to identify immune signatures and pathway hallmarks between NAFLD and normal participants, which would prepare us to construct co-expression networks among immune cells, signaling pathways, differential genes and transcription factors.

## Construction of regulatory network and screening of hub genes based on DEGs, DETFs, immune signatures and pathway hallmarks

In the previous study herein, we obtained results for DEGs, DETFs, immune signatures, and pathway hallmarks. On this basis, we used DEGs to perform correlation analysis (correlation more than 0.85; significance less than 0.05) with transcription factor/immune signatures/pathway hallmarks, respectively (Fig. 5A). The common genes of DEG, DETF, immune signatures and pathway hallmarks were screened and visualized by heat map (Fig. 5B). Then, we constructed the correlation coefficient matrix of common genes, immune signatures and pathway hallmarks, which were shown to be highly correlated with each other (Fig. 5C).

Next, to explore the hub genes in common genes, we used correlation coefficient as links and common genes as nodes to construct the co-expression network of DEGs, DETFs, immune signatures and pathway hallmarks. As shown in the network, *GTF2I* and *CHD2* were identified as hub genes between NAFLD and normal participants (Fig. 5D). We would validate the hub genes to clarify its function in the next step.

## The expression of *GTF2I* is higher in patients with NAFLD than healthy people

Free fatty acids (FFA, 1mM) were used to stimulate HepG2 cells for 24 h to construct an *in vitro* model of NAFLD. Oil red O staining confirmed that the lipid droplets increased significantly under the stimulation of FFA (Fig. S1). Meanwhile, the mRNA and protein

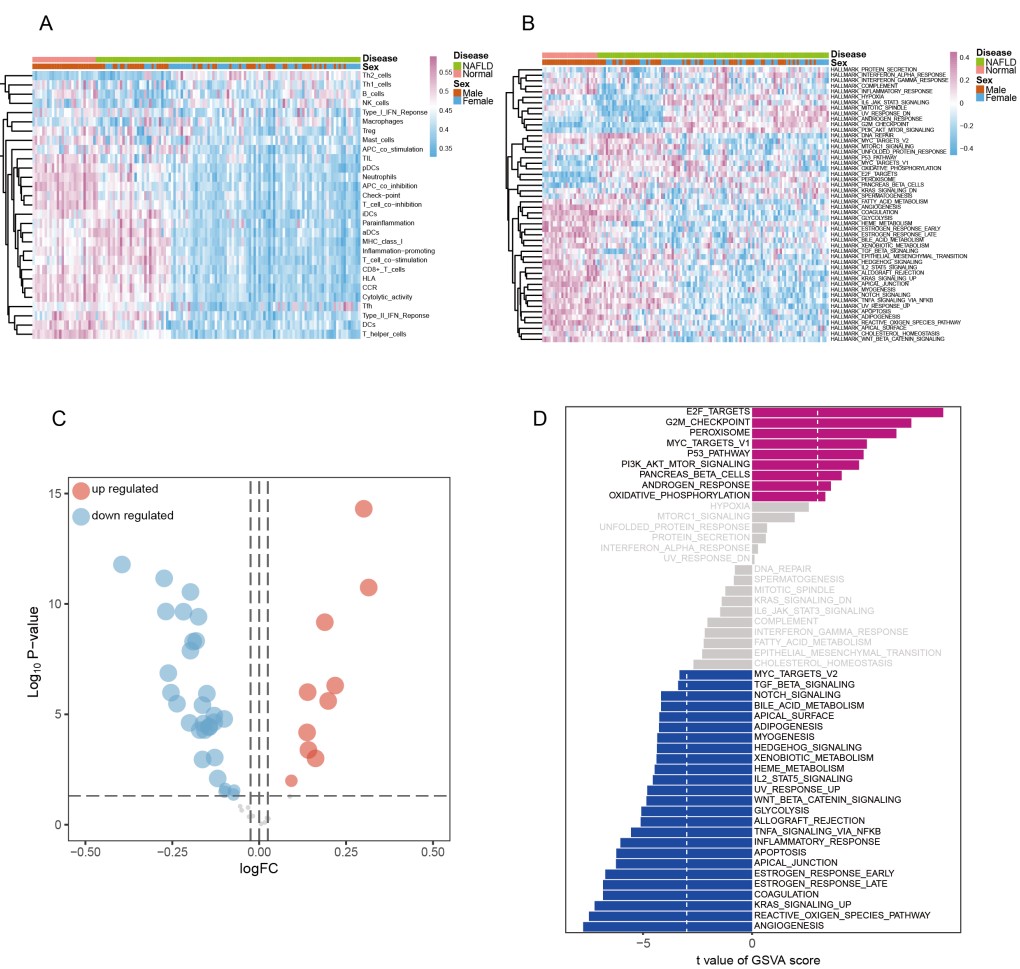

**Figure 4 Identification of Immune signatures and pathway hallmarks between NAFLD and normal participants.** (A) The immune signatures of 29 immune-associated gene sets were analyzed by GSEA and were visualized by heat map with unsupervised clustering. (B) The pathway hallmarks of 50 hallmark sets were analyzed by GSVA and were visualized by heat map with unsupervised clustering. (C–D) The significant pathway hallmarks were shown by volcano map and bar plot. Red bar indicates a positive correlation, and blue bar indicates a negative correlation.

expression of *GTF2I* were significantly increased in the FFA group compared with the control group. Conversely, the expression level of *CHD2* was not significantly changed by FFA stimulation (Figs. 6A–6B).

Next, a total of 13 histologically validated human samples (Normal =3; NAFLD =10) were used to examine the expression levels of *GTF2I* and *CHD2* (Fig. 6C). The results showed that the mRNA and protein expression levels of *GTF2I* in NAFLD patients were significantly higher than those in healthy people (Figs. 6D–6E). In contrast, the mRNA expression level of *CHD2* in NAFLD patients was significantly higher, while its protein expression was not detected (Figs. 6D–6E). Furthermore, IHC was also used to verify the relationship of *GTF2I* and NAFLD in human liver tissue samples. Compared with healthy individuals, *GTF2I* was significantly upregulated in NAFLD patients and appeared positive

**Table 2  The result of ssGSEA.**

|  | logFC | AveExpr | t | P.Value | adj.P.Val | B |
|---|---|---|---|---|---|---|
| APC_co_inhibition | −0.150226751 | 0.417994299 | −13.2383464 | 3.57E−26 | 1.03E−24 | 48.81073912 |
| DCs | −0.143566147 | 0.235410253 | −10.38211186 | 6.26E−19 | 4.54E−18 | 32.21322442 |
| Cytolytic_activity | −0.274620048 | 0.484078518 | −9.589045688 | 6.23E−17 | 3.01E−16 | 27.64384486 |
| T_helper_cells | −0.151201058 | 0.863347164 | −9.045849038 | 1.41E−15 | 5.83E−15 | 24.55271794 |
| CD8+_T_cells | −0.156963264 | 0.337882253 | −6.570642478 | 9.96E−10 | 2.41E−09 | 11.24603267 |

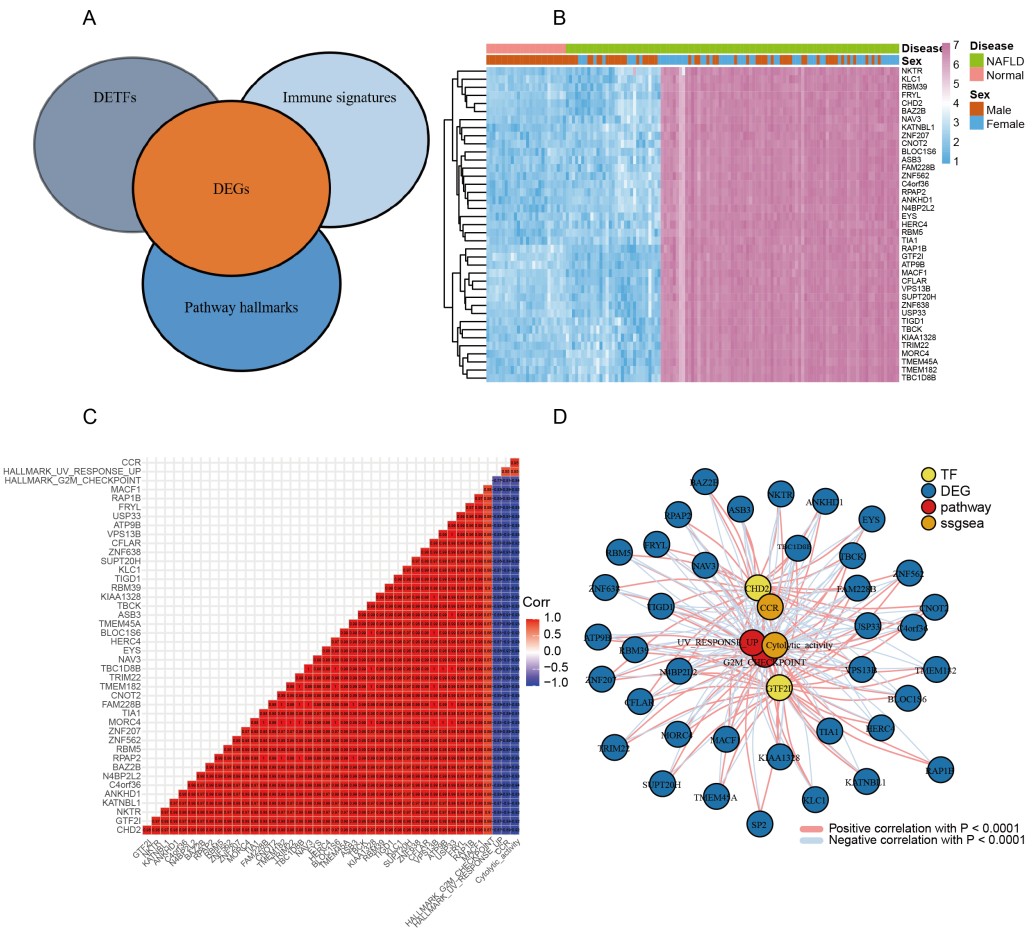

**Figure 5  Construction of regulatory network and screening of hub genes based on DEGs, DETFs, immune signatures and pathway hallmarks.** (A) Schematic diagram of correlation analysis between DEGs and DETFs/immune signatures/pathway hallmarks. (B) The common genes of DEGs, DETFs, immune signatures and pathway hallmarks were visualized by heat map with unsupervised clustering. (C) The correlation coefficient matrix of common genes, immune signature and pathway hallmarks. Red squares represent a positive correlation, and blue Squares represent a negative correlation. (D) The regulatory network of hub genes based on common genes, immune signatures and pathway hallmarks. Red lines represent a positive correlation, and blue lines represent a negative correlation.

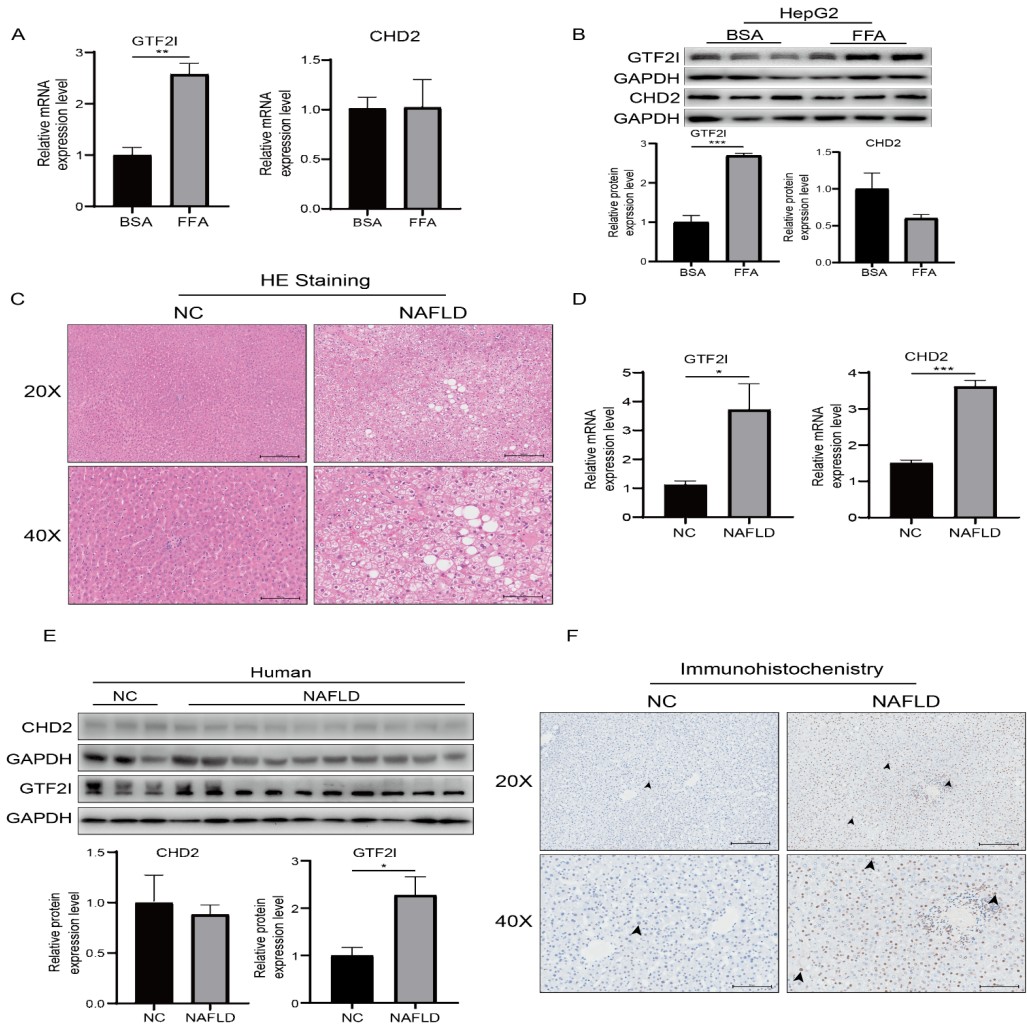

**Figure 6** **The mRNA and protein expression of *GTF2I* and *CHD2* in cell model and human samples.**
(A–B) The mRNA and protein of *GTF2I* was high expression in the FFA group, while the difference in
CHD2 was not obvious. (C) H & E staining was used as pathological evidence for the diagnosis of NAFLD.
(D–E) RT-qPCR revealed the mRNA expression of both *GTF2I* and *CHD2* were significantly higher in
NAFLD patients than in normal participants. (F) WB showed that *GTF2I* but not CHD2 was significantly
higher in NAFLD patients than in normal participants. (G) The immunohistochemical results of *GTF2I* in
liver tissues of NAFLD and normal participants. Arrows served as a marker of positive areas. * $P < 0.05$, **
$P < 0.01$, *** $P < 0.001$.

in the nucleus (Fig. 6F). These results suggested that *GTF2I* rather than *CHD2* may be
playing an important role in NAFLD.

## *GTF2I* may be involved in the regulation of immune profile by regulating chemokine receptor family

Based on bioinformatics analysis and experimental validation, we screened out that *GTF2I*
might be an important molecule in NAFLD. We further explored the relationship of *GTF2I*
and the immune profile.

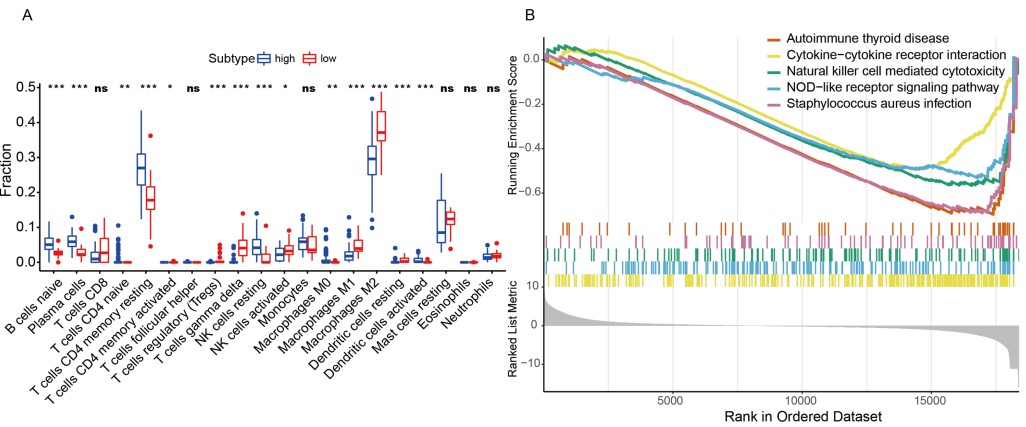

**Figure 7 Evaluation of the immune profile.** (A) Box plot showed the fraction of different immune cells from high expression and low expression group by non-parametric tests. Blue represents the high expression group, and red represents the low expression group. (B) High expression and low expression groups were differentially expressed in immune related pathway by GSEA.

  First, based on the NAFLD patient global expression matrix, we re-divided the high and low expression groups according to the expression level of *GTF2I* to clarify the effect of *GTF2I* on different immune cells. Interestingly, high expression of *GTF2I* group (NAFLD) was all NAFLD patients, while the *GTF2I* low expression group (Normal) was all non-NAFLD patients. CIBERSORT analysis revealed that *GTF2I* has a significant effect on the proportion of immune cells, including NK cells, CD4 + T cells, macrophages, and dendritic cells (Fig. 7A). On this basis, we further analyze the relationship of *GTF2I* and immune cell surface markers to clarify the specific mechanism of *GTF2I* in influencing the composition of immune cells. Surprisingly, *GTF2I* all affected the expression of surface molecules on NK cells, CD4 + T cells, macrophages, and dendritic cells (Fig. S2). This suggested that *GTF2I* might play core role in the regulation of immune cells. GSEA analysis showed that *GTF2I* is negatively correlated with 'Cytokine–cytokine receiver interaction' (Fig. 7B). Consistent with this, the correlation coefficient matrix and co-regulatory network showed that *GTF2I* is negatively correlated with chemokine receptors (CCR) (Fig. 5C). These results suggested that *GTF2I* may affect the immune profile in the liver by regulating the chemokine receptor family. Along this line, we compared the expression of chemokine receptor family between the two groups. As expected, *GTF2I* had a significant impact on the chemokine receptor family (227/261, count of significance) (Fig. 8), which may be the key pathways of *GTF2I* and need to be further confirmed in future work.

## DISCUSSION

NAFLD is a prevalent disease in the population that potentially seriously damages liver function, especially after progression to NASH, and the disease may further develop to liver fibrosis, cirrhosis, and even liver cancer (*Ye et al., 2020*). NAFLD not only affects human health, but the consequent medical costs and economic losses have also become a significant burden on patients (*Zhou et al., 2020*).

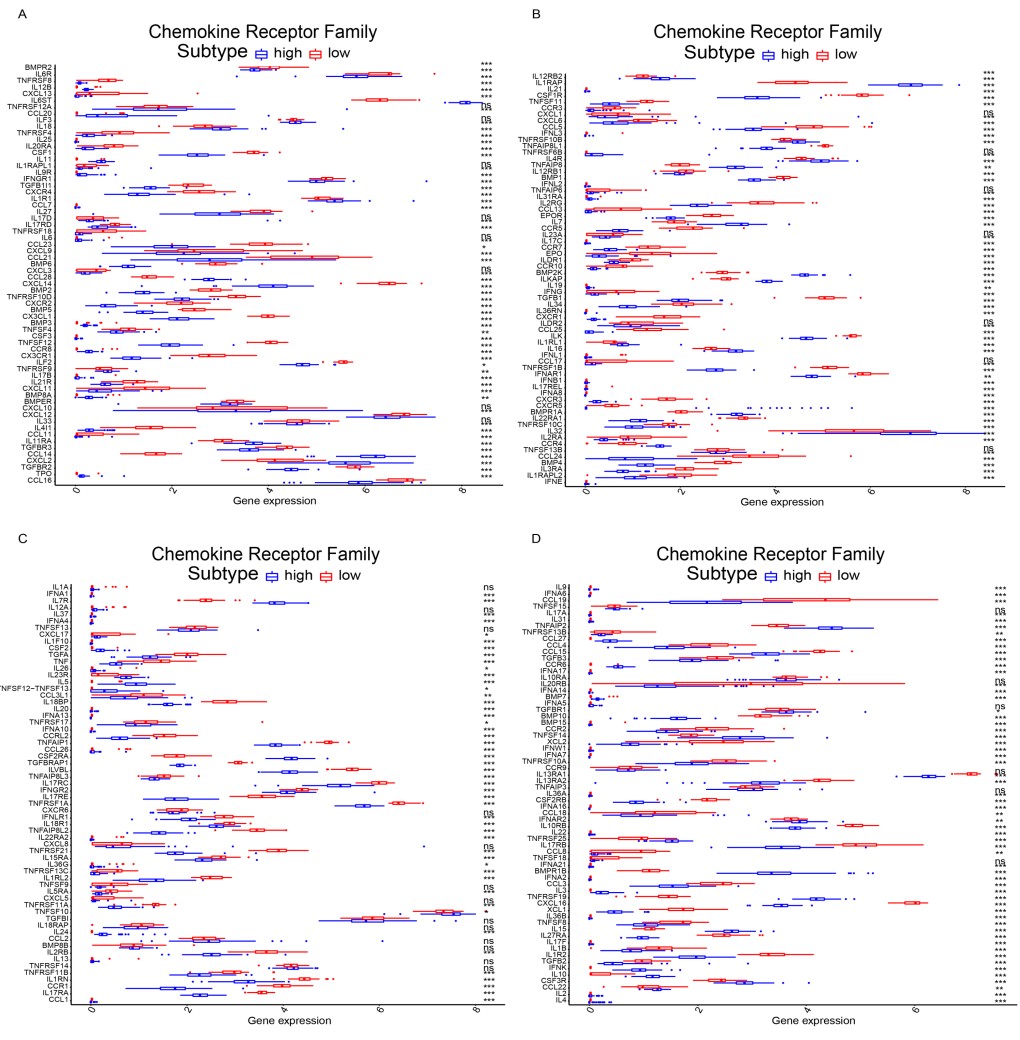

**Figure 8** **Correlation analysis of *GTF2I* and chemokine receptor family.** (A–D) In the high expression group and low expression group, a total of 261 chemokine receptor family molecules were compared, of which 227 were significant.

Currently, the pathogenesis of NAFLD mainly involves liver damage induced by multiple hits on the basis of ectopic accumulation of fat, including oxidative stress damage, inflammatory response, bile acid metabolism, lipotoxicity and endoplasmic reticulum stress (*Friedman et al., 2018*). Several studies involving transcription factors have shown therapeutic effects in NAFLD. For example, obeticholic acid (OCA), an *FXR* agonist, has been used clinically to treat NAFLD. Mechanistically, it can activate *FXR* to suppress bile acid production, thereby playing a therapeutic role in NAFLD (*Radun & Trauner, 2021*). Also, saroglitazar magnesium, a *PPAR* agonist, which can simultaneously regulate *PPARα* And *PPARγ*. It has therapeutic effects in both lipid metabolism and insulin resistance (*Gross et al., 2017*; *Wagner & Wagner, 2020*; *Lefere et al., 2020*). However, the effects of drugs targeting these TFs in clinical trials are limited, and some of the drugs often produce
serious side effects (*Bushweller, 2019*). Recently, the regulation of immune cells has received increasing attention in the germination and development of NAFLD. Although *PPARs* are mainly known for their roles in modulating lipid metabolism, *PPARγ* and *PPARδ* also have anti-inflammatory actions on macrophages, which may contribute to the potential clinical benefit in NASH (*Gallardo-Soler et al., 2008*; *Odegaard et al., 2007*; *Gordon, 2003*). Similar to the *PPARs, FXR* not only regulates lipid and glucose homeostasis but also exerts anti-inflammatory effects through inhibition of the recruitment of *NF-kB* on the promoter of several pro-inflammatory genes and stabilization of the binding of *NCor1* complex on the promoter of pro-inflammatory genes (*McMahan et al., 2013*; *Wang et al., 2008*). In addition, Liver X Receptors (*LXRs*) which are now recognized to be central regulators of cholesterol metabolism in mammals could regulate the expression of a panel of genes involved in reverse cholesterol transport in macrophages (*Mitro et al., 2007*; *Ogawa et al., 2005*). Therefore, the identification of an immune related transcription factor may facilitate the discovery of novel therapeutic targets in NAFLD.

In the present study, we spread out our work from four aspects as follows. Notably, patients included in this study had liver pathology ranging from simple fat to fibrosis and a disease spectrum ranging from steatosis to cirrhosis. The aim of our analysis was to find a wide variety of molecules. Thus, for early-stage NAFLD such as simple steatosis, the global gene expression profile is similar to that of normal participants. There were some patients with NAFLD whose heatmaps were similar to those of normal participants. Based on the gene expression information of NAFLD and normal participants, pathway hallmarks, immune signatures, DETFs and DEGs in NAFLD were screened out, respectively. To identify TFs, we performed a selection on the basis of DEGs in order to pick out the most weighted molecules. Transcription factors play a central role in signal transduction and growth metabolic processes (*Lambert et al., 2018*). Therefore, 50 canonical pathway information was used as a reference, and GSVA algorithm was used to facilitate the identification of important biological processes in NAFLD. In the identification of immune signatures, combining the reference information of 29 immune cells and pathways, we performed ssGSEA analysis on each sample to screen out common immune signatures in NAFLD. Then, how to integrate these results in order to explore their common characteristics?

First, we used DEGs to perform correlation analysis (correlation more than 0.85; significance less than 0.05) with transcription factor/immune signatures/pathway hallmarks, respectively. Genes that were present in both DEGs, DETFs, ssGSEA and GSVA were filtered out to be the final set of genes for analysis. Next, we used correlation coefficient as links and differential genes as nodes to construct the co-expression network of differential genes, transcription factors, immune characteristics and pathways. *GTF2I* and *CHD2* were identified as hub genes in the network. In this coregulatory network, we also focus on the cellular communication, especially chemokine receptor family, which play an important role in this network. This coregulatory network also provided direction for our subsequent analysis. Then, in terms of validation, we confirmed through human samples and cell models that *GTF2I*, but not *CHD2*, plays an important role in NAFLD. Meanwhile, we would in turn re-divide into high expression group and low expression group according

to the expression level of *GTF2I*. Among them, the high expression group was all NAFLD patients, while the low expression group was healthy participants, as reflected by the good discrimination ability of *GTF2I* in NAFLD.

The expression level of *GTF2I* significantly affected the composition of immune cells, including NK cells, T cells, dendritic cells, and macrophages. While combining the coregulatory network and GSEA results, we speculated that *GTF2I* might modulate the effects of multiple immune cells globally by negatively regulating chemokines. In this work, we constructed a co-regulation network by integrating diverse outcomes from DEGs, DETFs, immune signatures and pathway hallmarks. After experimental and bioinformatics reanalysis, a valuable molecule was successfully screened out. *GTF2I* was located at the center of the entire coregulatory network while exhibiting significant high expression in NAFLD cell models and NAFLD human samples. In addition, it also plays an important role in the immune profile by analyzing the effect of its expression level on immune cell composition. In addition, similar to the results of our analysis, studies from other investigators have also shown that *GTF2I* is associated with heavy chain immunoglobulin transcription in immune cells, which provides a reference and evidence for our study (*Gebru et al., 2021*). Through this work, we believe that *GTF2I* is an important molecule in NAFLD. The present analysis will provide a theoretical basis and experimental direction for us to pursue more levels of validation around *GTF2I* in the future.

## CONCLUSIONS

In conclusion, this research illustrated *GTF2I* was related to the pathogenesis and development of NAFLD. Mechanistically, *GTF2I* was identified to play a role in NAFLD by influencing the immune profile through negative regulation of chemokine receptor family. This study may provide a potential target for the diagnosis or therapy of NAFLD.

## ACKNOWLEDGEMENTS

We appreciated the authors, who provided these data series, and the authors of the databases used in this article. They completed this research with great efforts.

### Funding

This study was supported by funds from the National Natural Science Foundation of China (grant numbers: #81873567 to Wenzhuo Yang), Municipal Health Commission of Shanghai (NO. 81873567 to Wenzhuo Yang). The funders had no role in study design, data collection and analysis, decision to publish, or preparation of the manuscript.

### Grant Disclosures

The following grant information was disclosed by the authors:
National Natural Science Foundation of China: #81873567.
Municipal Health Commission of Shanghai: 81873567.

## Competing Interests

The authors declare there are no competing interests.

## Author Contributions

- Minbo Zhang conceived and designed the experiments, performed the experiments, analyzed the data, prepared figures and/or tables, and approved the final draft.
- Yu Zhang conceived and designed the experiments, performed the experiments, analyzed the data, prepared figures and/or tables, and approved the final draft.
- Xiaoxiao Jiao analyzed the data, prepared figures and/or tables, and approved the final draft.
- Linying Lai analyzed the data, prepared figures and/or tables, and approved the final draft.
- Yiting Qian analyzed the data, prepared figures and/or tables, and approved the final draft.
- Bo Sun analyzed the data, authored or reviewed drafts of the article, and approved the final draft.
- Wenzhuo Yang conceived and designed the experiments, authored or reviewed drafts of the article, and approved the final draft.

## Human Ethics

The following information was supplied relating to ethical approvals (*i.e.*, approving body and any reference numbers):

The Research Ethics Committee of Tongji Hospital approval to carry out the study within its facilities (Ethical approval number: k-kysb-2020-139).

## Data Availability

The raw data is available in the Supplemental Files.

## Supplemental Information

Supplemental information for this article can be found online at http://dx.doi.org/10.7717/peerj.13735#supplemental-information.

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
