# Peer review of "Identification and validation of immune related core transcription factors GTF2I in NAFLD"

_PeerJ, doi:10.7717/peerj.13735_

## Round 0.1 · original submission · Major Revisions

I would also like to recommend that -

(i) ALL of the reviewers' concerns are addressed in the revised manuscript.

(ii) A more detailed account of the results is provided in the results section.

(iii) A thorough grammatical and typographical editing of the manuscript is performed before resubmission.

·

Basic reporting

The authors analyzed publicly available transcriptome datasets of NAFLD and identified GTF2I as a transcription factor associated with NAFLD. NAFLD is a common chronic liver disease. Understanding its biology is important for researchers and clinicians to develop new therapies and treat patients. The manuscript is properly written in a standard scientific format. All results are relevant to the main topic.

There is a MINOR point the author needs to revise:
In line 77: However, there are only a few studies on the involvement of TFs in 78 NAFLD by regulating immune cells.
The authors need to provide references for these "a few studies".

Experimental design

The experiments in the manuscript are well designed. Proper standards have been applied to analyses in the study. All methods are provided for other researchers to replicate this study,

However, there are two points that the author needs to clarify:
1. Since the author used previously published datasets for this study, the criteria for sample inclusion or exclusion need to be stated in the manuscript. This will also help the authors to rule out irrelevant information about patients that may contribute to gene expression. For example, are patients with a smoking history included?

2. All figures with heatmap need to state whether the clustering is using a supervised or unsupervised method.

Validity of the findings

All the data in the paper has been provided in a correct format with proper statistical analyses.

However, there is a MAJOR inconsistency between the title and the conclusion:
The title demonstrated that GTF2I is involved in regulating the activation of NK cells. However, this conclusion is from a bioinformatic prediction but not an experimental validation. Thus, the title is an overstated conclusion.

The authors should either provide NK cell activation markers staining in GTF2I high and low samples or clearly state that GTF2I-regulated NK cell activation is a bioinformatics-inferred conclusion in the title.

Reviewer 2 ·

Basic reporting

The English of your manuscript must be improved before resubmission. I strongly suggest that you obtain assistance from a colleague who is well-versed in English.

Experimental design

The organization and subsections are also appropriate.
The survey methodology is consistent with comprehensive and unbiased coverage of the subject.
But some parts are confusing, such as 2. In lines 246 to 257, It is hard to understand what Figures 6A-C are. the figure legend and the context is a little different.

Validity of the findings

Technically solid, with no major concerns with respect to evaluation, resources, reproducibility, ethical considerations

Additional comments

In this study, based on the comprehensive bioinformatics analysis of healthy volunteers and 109 NAFLD patients’ liver samples, 39 DEGs of liver samples between NAFLD patients and healthy volunteers were identified. Furthermore, the authors identified two key DETFs from the 39 DEGs, both GTF2I and CHD2. To further verify whether the identified transcription factors GTF2I and CHD2 were correct, the authors performed western blot, qPCR, and immunohistochemical experiments and found that GTF2I was significantly higher in NAFLD tissue samples and no significant difference in CHD2 expression.

1. In this study, the authors identified 39 differentially expressed genes, why focus on these two genes, GTF2I and CHD2?
2. Figures 6a and 6d both show HE staining but are completely different.
3. Figure 6G, In the right Panel (NAFLD) the scale is missing, please add. Please consider adding an arrow indicating the positive GTF21 in the right lower corner.
4. The author showed GTF21 may be involved in NAFLD by regulating the proportion of activation of NK cells. It would be better if the authors could co-stain GTF21 with NK cell makers or immune cell markers.
5. The primer sequence is confusing, please show it in the three-line table (do not draw vertical lines, horizontal lines only retain the top line, bottom line, and column line).

·

Basic reporting

The manuscript by Zhang M, and Zhang Y et.al has efficiently utilized bioinformatics and cell biology tools to identify a key transcription factor GTF2I associated with NK cells and highlight its putative role in NAFLD disease pathogenesis. The finding is quite interesting and based on the extensive analysis of a large dataset of 109 NAFLD patients.
The reporting of the manuscript is quite simple and conceptually easy to follow. The structure of the manuscript conforms to PeerJ guidelines. However, there is a need to improve the overall language of the manuscript to make it more understandable. There are few sentences which are incomplete, and there are quite a few grammatical and typographical errors throughout the manuscript. Certain words can be replaced to bring out the meaning clearly. I would suggest check for appropriate use of uppercase and lowercase letters where applicable. It is highly advised to take help of some appropriate online scientific editing and proofreading software, or the help of a colleague who is proficient in English to proofread the manuscript so that it is easy to understand by the readers. I have added few major edits in the additional comment section for the authors to re-check.
The manuscript is well-referenced throughout, and the references are correct. For Line 72 and 73 - For loss of the hepatic TF ChREBP in aggravating hepatic steatosis and insulin resistance, please cite the original articles instead of the review article.
The overall quality of the figures needs to be improved in resolution especially for Figure 2 and 4. Some of the color schemes needs to be changed (for example in Fig 4D) to enable reading the annotations. Some of the figures are not properly referenced or explained in the manuscript, which needs to be rechecked. I have elaborated more about results and figures in the Additional Comments section.
I thank the authors for sharing the Raw Data associated with the study. Only the Raw data for mRNA expression of CHD2 in BSA and FFA treated HepG2 cells is missing and should be added.

Experimental design

The experimental design of the study is scientifically sound and commendable and is based on extensive analysis of the public databases. The authors have also validated the bioinformatic finding nicely with in-vitro experiments. However, the manuscript would highly benefit from elaborate explanation of the figures in the results section and in the legends. The methods have been mostly well explained in detail. The dose/ concentration of FFA should be mentioned in the methods and the figure, and the reason for selecting that particular dose should be discussed or cited from literature.

Validity of the findings

The authors should explain the findings of the experiments as be lucidly and clearly as possible in the results section, with proper figure references. The authors have focused only on NK cells as a probable pathogenic immune cell for NAFLD, while their data also suggests possible role for T cells which also has high GTF2I expression.

Additional comments

There are several key concerns in the result and discussion section:
• Figure1: Typographical error of “Gene Ontology”, can consider replacing “fraction of immune cells” with “immune cell subset/population”
• Figure 2a and 3a) The color code legend in volcano plot to indicate healthy and NAFLD group is missing.
• Figure 2b 3b and 5a) It will be useful to highlight (with a box) the DEGs and DETFs which were considered for downstream analysis. There is a nice and clear distinction between healthy and most of NAFLD patients for DEGs and DETFs. However, a part of the NAFLD patients is very similar in genes expression and transcription factors profile to the healthy group. Is there a subgroup within the NAFLD patient dataset?
• Figure 2C-F: The annotations are missing words in all the figures. Especially in Figure 2F, it is not clear how the genes are associated with Huntington disease, Parkinson disease etc. because these diseases are not mentioned in the plot. I would suggest improving quality of the figures and adding legends wherever applicable.
• Figure 3A and Figure 3B labelled wrongly in the manuscript, and Figure 3C is missing. Kindly provide the full form of TSD atleast once in the manuscript. There is lack of explanation of why T cells were not explored further although there were obvious differences in T cell subsets between NAFLD and non-NAFLD subjects.
• There is no explanation or reference in the manuscript for Figure 4A, B and 4D.
• It would be helpful to represent Figure 5B and 5D in a better way with clear explanation in the manuscript for easier understanding.
• Figure 6A has been improperly referenced in the manuscript and the figure legends. Figure 6A shows Oil red staining of NAFLD and Control tissues but has been mislabeled as HE staining. Figure 6D shows HE staining and should be properly referenced in the manuscript.
• Kindly label the 2 plots of Figure 6B to indicate which cells (HepG2/human liver tissue) were analyzed for GTF2I in the 2 plots.
• The statistical analysis in Fig 7A needs to be clearly represented for better understanding. The authors can consider breaking up the figure into 2 parts so that the figure is clear. A key observation is that memory T cells also shows high expression of GTF2I in NAFLD group, and maybe expressed more than NK cells. The authors should explain in the results and the discussion, why the role of T cells was not explored further.
• The probable GTF2I involvement in ER stress is an interesting proposition and can be followed-up in future studies by the group. However, I do feel that the ER stress biology has been unnecessarily elaborated in the discussion and can be reduced. The possibility of T cells and macrophages association with NAFLD could be discussed based on the high expression of GTF2I in these cells, and also based on knowledge in current literatures.
• An interesting observation is that free-fatty acids induces expression of GTF2I in HepG2 cells. The authors may discuss the probable mechanism of GTF2I upregulation on FFA exposure of cells in terms of what molecules/cell surface receptors/transcription factors might control GTF2I expression in liver cells and immune cells. This can also pave way for further analyzing the dataset for these GTF2I inducing markers.

---

## Round 0.2 · accepted · Accept

All the reviewers' comments have been addressed.

·

Basic reporting

The author clearly addressed the question from a reviewer.

One more thing needs to be double-checked:
In general, gene symbols need to be italicized while protein names do not. Since most data of the studies are from RNA-seq, the gene mentioned should be italicized.

Experimental design

No comments

Validity of the findings

The author has addressed all the questions from a reviewer.

Reviewer 2 ·

Basic reporting

N/A

Experimental design

N/A

Validity of the findings

N/A

Additional comments

N/A

·

Basic reporting

The study has significantly improved after revision in terms of English language use and proficiency. The study is meaningful, well-structured and easy to follow. The conclusions have been explained sufficiently. The authors have added the necessary references as suggested in the previous comments. The quality of Figure 2 and 4 has been improved both in resolution and have been properly annotated and labelled.

Experimental design

The experimental design is good, and the figures have been properly reference in the manuscript as per suggestions. The dose of FFA has been added in the methods section and has been referenced. The methods have been appropriately discussed.

Validity of the findings

The results and conclusions have been logically explained. The authors have addressed the concern of focussing not only on NK cells, but also on other immune cells in the revised version

Additional comments

All the concerns raised previously have been addressed satisfactorily by the authors. There are no additional comments, except suggesting another round of proofreading for fixing some minor errors. For example, in Figure 1: The 'L' in liver RNA-SEQ should be in uppercase.